# Stochastic Natural Vibration Analyses of Free-Form Shells

**Bingbing San [1], Yunlong Ma [1], Zhi Xiao [1], Dongming Feng [2],*** 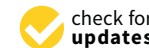 **and Liwei Yin [1]**

[1]   College of Civil and Transportation Engineering, Hohai University, Nanjing 210098, China
[2]   Thornton Tomasetti, Inc., Weidlinger Transportation Practice, 40 Wall St., New York, NY 10005, USA
*    Correspondence: df2465@columbia.edu

**Abstract:** This work investigates the natural vibration characteristics of free-form shells when considering the influence of uncertainties, including initial geometric imperfection, shell thickness deviation, and elastic modulus deviation. Herein, free-form shell models are generated while using a self-coded optimization algorithm. The Latin hypercube sampling (LHS) method is used to draw the samplings of uncertainties with respect to their stochastic probability models. ANSYS finite element (FE) software is adopted to analyze the natural vibration characteristics and compute the natural frequencies. The mean values, standard deviations, and cumulative distributions functions (CDFs) of the first three natural frequencies are obtained. The partial correlation coefficient is adopted to rank the significances of uncertainty factors. The study reveals that, for the free-form shells that were investigated in this study, the natural frequencies is a random quantity with a normal distribution; elastic modulus deviation imposes the greatest effect on natural frequencies; shell thickness ranks the second; geometrical imperfection ranks the last, with a much lower weight than the other two factors, which illustrates that the shape of the studied free-form shells is robust in term of natural vibration characteristics; when the supported edges are fixed during the shape optimization, the stochastic characteristics do not significantly change during the shape optimization process.

**Keywords:** free-form shells; natural vibration; stochastic analysis; sensitivity analysis

## 1. Introduction

The term free-form shells refers to shells whose geometric shapes cannot be represented by certain mathematical formulas or their combination [1]. When compared with traditional shells, such as cylindrical shells and spherical shells, the geometrical shape of free-from shells is flexible to satisfy certain structural, architectural, and other requirements, which advances the wide application of free-form shells in civil engineering, aerospace engineering, and so on.

The mechanical properties of free-form shells are strongly dependent on their geometric shapes. Therefore, shape optimization became an indispensable approach for designing a free-form shell. Bletzinger et al. [2] optimized free-form shells for structural stiffness under given loads while using numerical methods, which were emerged with physical experiments. Ohmori and Hamada [3] combined Non-Uniform Rational B-Spline (NURBS) with genetic algorithm (GA), as well as the gradient method to perform shape optimization of free-from shells, in which the objective functions are set as minimum strain energy and minimum geometric deviation from the prescribed shape. Cui and Yan [4] developed a height adjusting method to obtain optimal shape of free-form shells, with respect to minimum strain energy. Feng and Ge [1] described a conjugate gradient optimization method for the shape design of cable-braced free-form grid shells, with strain energy as the optimization objective. Wang and Wu [5] conducted a study on local optimal solutions and a modified optimization method, which was applied on the shape optimization of cable-stiffened latticed free-form shells, with the

objective of the minimization of strain energy. It can be seen that, in the past decades, numerical optimization methods have been developed for free-form shells, many of which made efforts to obtain a geometrical shape with minimum strain energy, i.e., largest structural stiffness.

There is a relative dearth of research on mechanical properties of free-form shells despite the fast development of shape optimization of free-form shells. Only minimal literature is concerned with that. For example, Li [6] studied the buckling properties of free-form shells; Cui et al. [7,8] carried out experimental and numerical studies on the static behaviors of the free-form concrete shells. However, the existing work mainly studies static characteristics and it is limited to the deterministic analysis without any consideration of uncertainty factors, which are unavoidable in practical engineering, such as geometrical imperfection and deviations of material properties. The previous stochastic studies on traditional shells have demonstrated the significant effect of uncertain factors on mechanical properties of structures [9–13]. The construction process of free-form shells is more complex when compared with traditional shells, which tends to lead more uncertainty factors. This further motivates this study to consider the effects of uncertainty factors on mechanical properties of free-form shells, particularly on the dynamic characteristics.

In this study, the stochastic natural vibration analyses of free-form shells is carried out while considering three uncertainty factors, including geometric imperfection, shell thickness deviation, and elastic modulus deviation. Free-form shell models are generated by a self-coded numerical optimization algorithm. Latin hypercube sampling (LHS) is used for the random sampling of three uncertainty factors based on their stochastic probability modeling. The global sensitivity study is conducted to rank the importance of these error sources.

The outline of this paper is as follows: in Section 2, the shape generation method of free-form shells is presented and programmed; Section 3 describes the stochastic and sensitivity analysis method; in Section 4, stochastic and sensitivity analyses are carried out on free-form shell, which are designed by the shape generation algorithm presented in Section 2. The results are reported along with discussions; finally, Section 5 provides closing remarks.

## 2. Shape Generation of Free-Form Shells

### 2.1. Shape Parametrization

The geometric shape of free-form shells cannot be represented by certain mathematical formulas or their combination, different from regular shells, such as spherical shells and cylindrical shells. Therefore, shape parameterization is needed to describe their arbitrary shapes with desired smoothness. NURBS is employed here to represent the shape of structures, which is developed from B-spline and have been widely applied. This section gives a brief introduction on NURBS. For more details, one can refer to [14].

A NURBS surface, as shown in Figure 1, is defined as

$$S(u,v) = \frac{\sum_{i=1}^{m}\sum_{j=1}^{n} P_{i,j}\omega_{i,j}N_{i,k}(u)N_{j,l}(v)}{\sum_{i=1}^{m}\sum_{j=1}^{n}\omega_{i,j}N_{i,k}(u)N_{j,l}(v)} \tag{1}$$

where $S(u,v)$ is the value of a point on the NURBS surface, $P_{i,j}$ represents the coordinate positions of a set of control points, which form a $m \times n$ bi-directional control-point grid, and $\omega_{i,j}$ is their respective weights; $N_{i,k}(u)$ and $N_{j,l}(v)$ are B-spline basis functions, defined on the knot vector $\Xi_u = \left\{ u_1, \quad u_2, \quad \cdots \quad u_{m+k+1} \right\}$ and $\Xi_v = \left\{ v_1, \quad v_2, \quad \cdots \quad v_{n+l+1} \right\}$, where $u_1 \leq u_2 \leq \cdots u_{m+k+1}$, $v_1 \leq v_2 \leq \cdots \leq v_{n+l+1}$; $u_i$ and $v_i$ are the real numbers representing the coordinates in the parametric space $[0,1]$; $k$ and $l$ are degrees.

$N_{i,k}(u)$ is defined by

$$N_{i,0}(u) = \begin{cases} 1 & u_i \le u < u_i + 1 \\ 0 & \text{otherwise} \end{cases}$$

$$N_{i,k}(u) = \frac{u-u_i}{u_{i+k}-u_i}N_{i,k-1}(u) + \frac{u_{i+k+1}-u}{u_{i+k+1}-u_{i+1}}N_{i+1,k-1}(u) \quad (i = 1, 2, 3 \cdots)$$

(2)

while $N_{j,l}(v)$ is defined in the similar way.

The aforementioned equations have revealed that the geometry of a NURBS surface is affected by its control points and their weights. However, the work that is presented in [6] shows that the control points play a dominant role in changing shapes rather than the weights. Although setting both locations and weights of control points as design variables could give more flexibility to the shape and enlarge the design space, it would significantly increase the computational cost. For this reason, we only consider the coordinates of control points as the optimization variables while keeping the weights fixed. Particularly, the optimal shape is approached by updating the Z-coordinate of the control points, while setting their X, Y-coordinates as the fixed parameters.

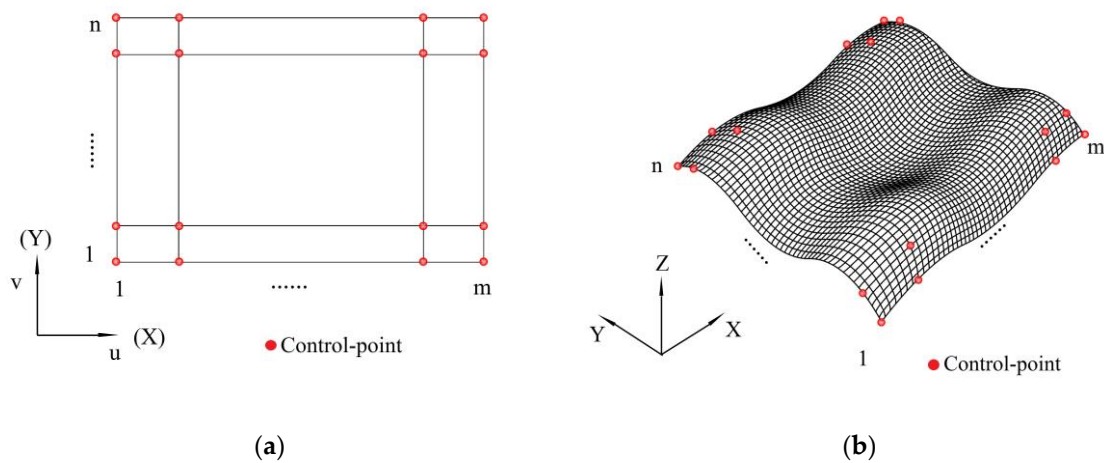

(**a**)                                                                                            (**b**)

**Figure 1.** Example of a NURBS (Non-Uniform Rational B-Spline) surface: (**a**) Control-point grid; (**b**) Corresponding NURBS surface.

### 2.2. Optimization Algorithm

In this study, the free-form shape is generated through an optimization approach that minimizes the strain energy. A gradient-based method is employed to solve the problem, in which a gradient descent, or negative gradient direction of the objective function is selected as the search direction of each iteration step, and it gradually approaches the minimum function value. In gradient-based optimization, it is indispensable to solve the derivatives of the objective and/or constraint functions with respect to the design variables, which could be computed while using analytical, semi-analytical, or numerical methods, depending on the problem itself and the available resources [15]. Here, a numerical differentiation method is selected to compute derivatives, since we have relatively small number of design variables.

Combining the finite element (FE) forward analysis with the shell shape being parameterized by NURBS, the gradient-based optimization procedure is obtained. The code is programmed while using the FORTRAN language. The FE modeling is based on the Kirchhoff–Love plate theory and the plane triangular shell element with 3 nodes is employed. For better understanding, the algorithm is described in Algorithm 1.

---

**Algorithm 1**: Shape generation of free-form shells

---

1. *Given material properties, shell thickness, load, boundary conditions*
2. *Set up the location of control points*
3. *Assign an initial guess for the design variables, i.e., Z-coordinates of control points*
4. *Select the convergence criteria, i.e., the relative change of the objective function is smaller than a given tolerance value*
5. **Loop until convergence**

    (1) *Implement NURBS to obtain the geometry of the shape*
    (2) *Generate FE mesh with surface parameterized by NURBS*
    (3) *Solve the FE forward problem and obtain the objective function value*
    (4) *Compute derivatives of objective with respect to $Z_i$ using a numerical method*
    (5) *Compute the step size of $Z_i$ by the Golden section method*
    (6) *Update the design variables $Z_i$*

6. **End loop**

---

## 3. Stochastic and Sensitivity Analysis Method

The natural vibration analyses are coupled with LHS to investigate the stochastic characteristics of natural frequencies of free-form shells and to explore the sensitivity of the uncertainty factors.

LHS [16–18] is a widely-used sampling method, which allows for the extraction of a large amount of stochastic and sensitivity information with a relatively small sample size. It can ensure that the entire distribution of each input random variable is covered and good at estimating the mean values of output variables [19]. LHS consists of two main steps, sampling and permutation. Sampling is implemented to produce representative samples to describe the distribution of each input random variable, while permutation aims at reducing the correlations between the samples of different random variables.

Herein, LHS is used for sampling. $e_1, e_2, \cdots, e_{n_e}$ denote the input uncertainty factors. Their respective probability cumulative function is expressed as $F_i = F_i(e_i)$. $f$ represents the output variable, i.e., the natural frequency in the current study. For a sample size $n_s$, the range of $F_i$ is divided into $n_s$ equal interval zones, which are non-overlapping. One value is randomly selected for each intervals and the sampling. A row of sampled values $e_{i1}, e_{i2}, \cdots, e_{in_s}$ is obtained. A $n_e \times n_s$ sampling matrix can be generated after all the input random variables are sampled. Subsequently, random permutation is performed, in which the sampled value of every input variable is randomly chosen out of the sample values from the corresponding row in sampling matrix without replacement.

To check the sample size, the convergence estimation formulas are given, as follows

$$check_1 = \left| \frac{\overline{f}(n_s) - \overline{f}(n_s - ch)}{\overline{f}(n_s)} \right| \leq \xi_1; \ check_2 = \left| \frac{\sigma(n_s) - \sigma(n_s - ch)}{\sigma(n_s)} \right| \leq \xi_2 \tag{3}$$

where $n_s$ is the number of samplings; $ch$ is added number of samplings; $\overline{f}(n_s - ch)$, $\overline{f}(n_s)$ is mean value of output results with $n_s - ch$ samplings and $n_s$ samplings, respectively; $\sigma(n_s - ch)$ and $\sigma(n_s)$ is standard deviation of output results with $n_s - ch$ samplings and $n_s$ samplings, respectively; referring the existing work [19–21], the tolerances $\xi_1$ and $\xi_2$ are determined as values that are less than or equal to 0.01.

Based on LHS, the natural frequencies of free-form shells, which involve inherent randomness, is investigated by a partial correlation-based method [17–19,22,23]. The partial correlation coefficient was proposed by Iman and Helton [22], which is used to measure the degree of the linear correlation between the input variable and the model output after making an adjustment to remove the linear effect of all the remaining variables. The derivation of the partial correlation coefficient can be referred to [23]. Herein, the formulation of the partial correlation coefficients between input uncertainty factors $e_i$ and output variables $f$ (i.e., natural frequencies in the current study) is given, as follows [22,23]



$$S_i = -c_{if} / \sqrt{c_{ii}c_{ff}} \tag{4}$$

where $c_{if}$, $c_{ii}$, and $c_{ff}$ are the elements of matrix $C$, written as follows

$$
C = \begin{bmatrix}
c_{11} & c_{12} & \cdots & c_{1n_e} & c_{1f} \\
\cdots & \cdots & \cdots & \cdots & \cdots \\
\cdots & \cdots & \cdots & \cdots & \cdots \\
c_{n_e1} & c_{n_e2} & \cdots & c_{n_en_e} & c_{n_ef} \\
c_{f1} & c_{f2} & \cdots & c_{fn_e} & c_{ff}
\end{bmatrix}
=
\begin{bmatrix}
r_{11} & r_{12} & \cdots & r_{1n_e} & r_{1f} \\
\cdots & \cdots & \cdots & \cdots & \cdots \\
\cdots & \cdots & \cdots & \cdots & \cdots \\
r_{n_e1} & r_{n_e2} & \cdots & r_{n_en_e} & r_{n_ef} \\
r_{f1} & r_{f2} & \cdots & r_{fn_e} & r_{ff}
\end{bmatrix}^{-1}
\tag{5}
$$

where $r_{ij}$ is linear correlation coefficient between input variables $e_i$ and $e_j$; $r_{if}$ is the linear correlation coefficient between input variable $e_i$ and output variable $f$, see Equation (6)

$$r_{if} = \frac{\sum\limits_{k=1}^{n}\left(x_{ik} - \overline{x}\right)\left(f_k - \overline{f}\right)}{\sqrt{\sum\limits_{k=1}^{n}(x_{ik} - \overline{x})^2}\sqrt{\sum\limits_{k=1}^{n}\left(f_k - \overline{f}\right)^2}} \tag{6}$$

Partial correlation coefficients $S_i$, are dimensionless parameters with a range between $-1.0$ and $+1.0$ and reveal the degree of association between the input and output variables. Herein, partial correlation coefficients are applied to rank the importance of each input variables.

In this study, three uncertainty sources $e_1, e_2, e_3$ are considered, which are initial geometric imperfection of shells, shell thickness deviation, and elastic modulus deviation. The output variable $f$, i.e., natural frequency, is computed by software ANSYS while using the Block lanczos method.

## 4. Results and Discussion

As presented above, three main stochastic factors that may occur on shells are explored, including geometric imperfection of shells, shell thickness deviation, and elastic modulus deviation. The distribution of initial geometric imperfection throughout the shell is determined by the consistent mode imperfection method [24], while the shell thickness deviation and elastic modulus deviation are assumed to be uniformly distributed over the shell. Since each of uncertainty error sources is determined by several independent factors, they are assumed to follow normal distribution [25]. Referring to the existing work and code [26,27], their respective mean values and standard deviations are determined, as shown in Table 1, where initial geometric imperfection is described by the maximum height deviation of the shell; the thickness and elastic modulus deviation is represented by the relative error between actual and design values, with "+" and "−" meaning the actual value larger and smaller than the design value, respectively.

**Table 1.** Stochastic models of uncertainties.

| Uncertainty Factors | Value Ranges (Guarantee Rate) | Mean Values | Standard Deviation |
|---|---|---|---|
| Initial geometric imperfection | −100 mm ~ + 100 mm (95.0%) | 0 | 51.0 mm |
| Thickness deviation | −3.57% ~ + 5.71% (95.0%) | 1.07% | 0.02369 |
| Elastic modulus deviation | −15.00% ~ + 15.00% (99.6%) | 0 | 0.05000 |

The presented stochastic analysis method is implemented on two widely used free-form shells. One is a free-form shell with negative Gaussian curvature and the other one is a combination of two positive-Gaussian domes. Both of the models are generated by the optimization approach, which is presented in Section 2. The natural vibration analysis is implement by ANSYS finite element software.

*4.1. Shell Model 1: A Negative-Gaussian Free-Form Shell*

4.1.1. Model Generation

The shell model in this example is as shown in Figure 2, which cover a rectangle plane section with a dimension of $8 \times 12m$. Both long-span edges are simply supported, while the two short-span edges are free. The material properties and the design shell thickness are as shown in Table 2. The material is assumed to be linear elastic.

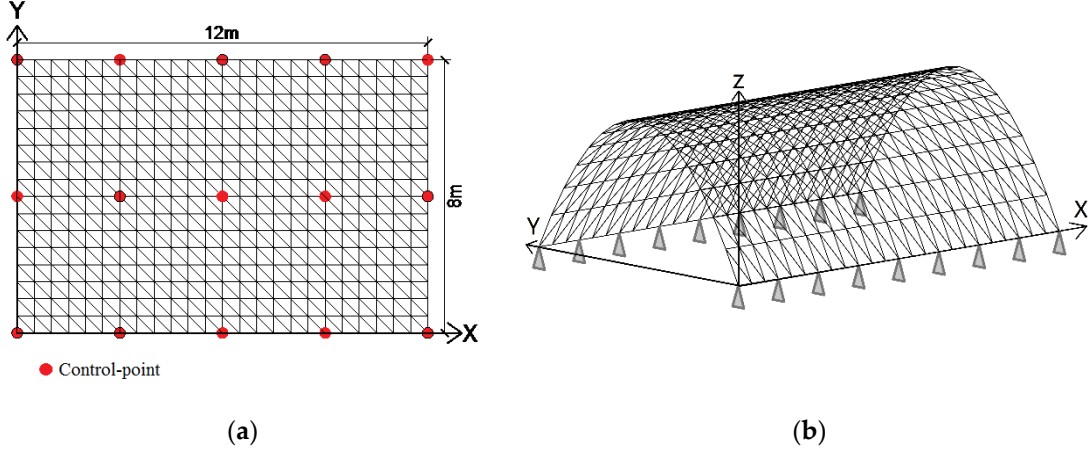

|            (a)            |            (b)            |

**Figure 2.** Layout of control points, mesh and initial guess shape of model 1: (**a**) Layout of control points and mesh, (**b**) Initial guess shape.

**Table 2.** Material parameters and shell thickness.

| Elastic Modulus (MPa) | Poisson's Ratio | Thickness (mm) | Density (kg/m$^3$) |
| :---: | :---: | :---: | :---: |
| $3.0 \times 10^4$ | 0.617 | 140 | 2500 |

The shape is generated by the proposed optimization approach. NURBS functions are adopted to represent the geometric shape, with control points that are shown in Figure 2a, which forms a $5 \times 3$ grid. As aforementioned, the weight factors have small effects on the NURBS surface [6]. Therefore, the weight factors of all control points are chosen randomly from the range 0~1.0 as 0.5 and not changed during the optimization process. Both the degrees of the basis functions in the two directions are selected to be 2. The optimization aims at minimizing the strain energy under a given uniformly distributed vertical load $q = 3.5\text{kN}/\text{m}^2$. The strain energy is solved by a self-coded program, in which triangular shell element is used with 768 elements and 425 nodes.

The gradient-based algorithm is used to solve the optimization problem. A cylindrical shell with the height of 4 m is given as an initial guess, as shown in Figure 2b. In this example, the control point on the long-span edges are fixed meaning the location and shape of the two edges are not changing during the optimization process. Consequently, the total number of optimization variables is 5.

The variation of the objective function during the optimization process is depicted in Figure 3 to show the convergence process, in which the vertical coordinate indicates the ratio of the strain energy of the $i$th optimization step to that of the initial guess. It can be seen that the strain energy decreases significantly during the first 50 optimization steps. After the 50th step, the strain energy decreases slowly and then converges at the 400th step. Through the optimization, the strain energy decreases to 40% of that of the original design.

In Figure 4, the initial, 20th, 200th, and 400th (optimal) shapes are selected to show the shape variation during optimization process. It is observed that, via the first 20 optimization steps, the shape changes from a cylinder to a saddle to earn larger stiffness. Since the 20th step, the shape remains

saddle-shaped, and the surface curvature increases overall. The shape change is too slight to be observed after the 200th optimization step.

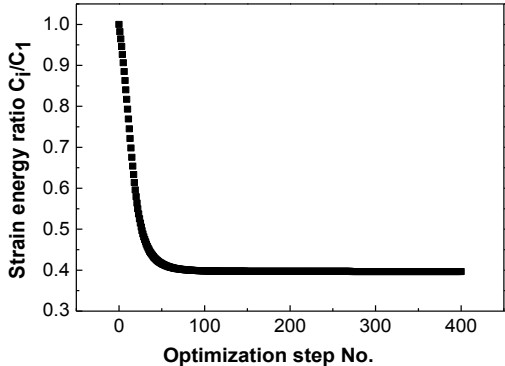

**Figure 3.** The convergence process of Model 1.

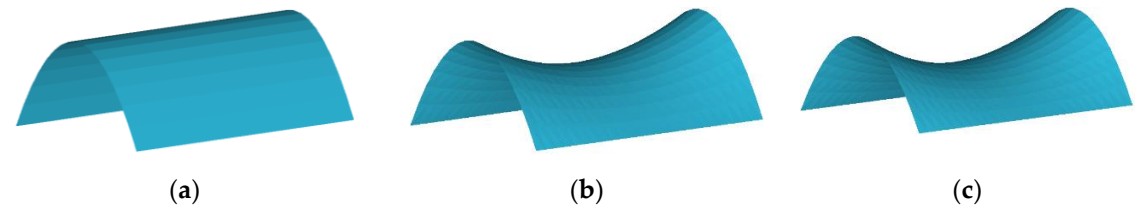

|      (a)      |      (b)      |      (c)      |

**Figure 4.** Shape variation of Model 1 during the optimization process: (**a**) Initial guess; (**b**) 20th step; and, (**c**) 200th/400th step (optimal).

Prior to the stochastic analysis, the deterministic analysis of free vibration is implemented on shells with respect to several selected optimization steps, and the first three natural frequencies are plotted in Figure 5. It can be seen that, all the first three natural frequencies significantly change during the optimization process and increase by 158.0% (1st), 61.1% (2nd), and 111.5% (3rd), respectively.

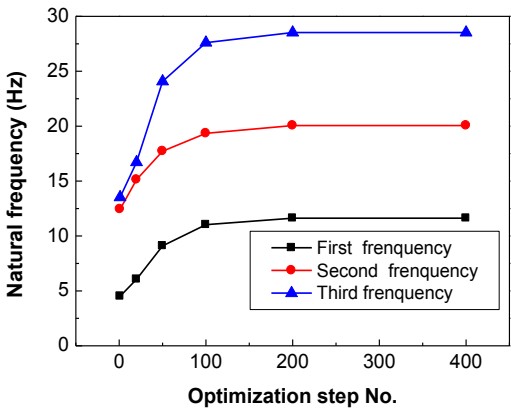

**Figure 5.** Evolution of the first three natural frequencies of Model 1 during the optimization process.

4.1.2. Stochastic and Sensitivity Analysis

Three representative shapes generated in the optimization, i.e., the initial shape, the 20th-step shape, the 400th-step optimal shape, see Figure 4, are selected for the stochastic analysis of natural frequencies.

LHS is adopted to generate samples. To determine the sample size, sampling is repeated on the optimal shape with size of 350, 400, 450, 500, 550, and 600, respectively. The mean values and standard deviations of first three frequencies are solved and checked by Equation (3), with $\xi_1$, $\xi_2$ set up as 0.01 and 0.001, respectively. It is found that, when the sampling size is 400, Equation (3) is satisfied for the

first three frequency of all the selected shapes. For brevity, only the check result of the first natural frequency of the optimal shape is given in Figure 6. Herein, the size of samples is determined as 400 and Figure 7 depicts the samples.

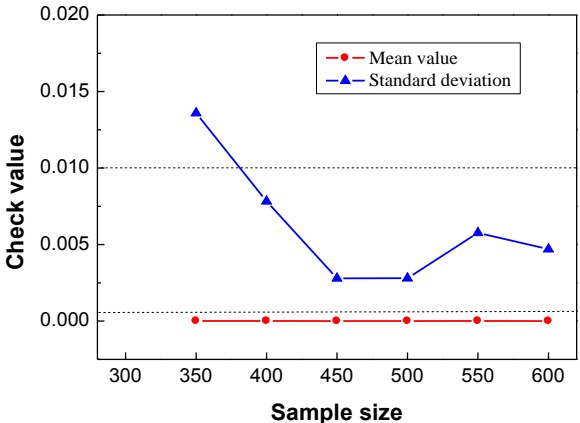

**Figure 6.** Sampling size test of first frequency every 50 samples.

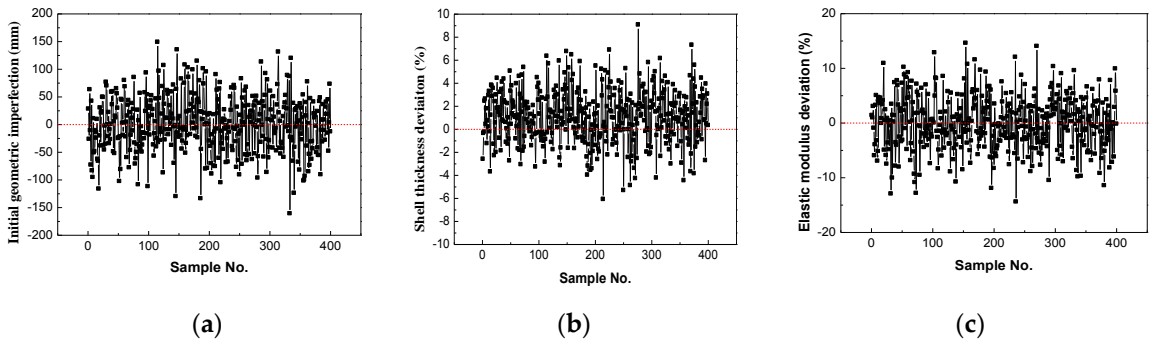

(**a**)  (**b**)  (**c**)

**Figure 7.** Samples of uncertainty parameters with size of 400: (**a**) Initial geometric imperfection; (**b**) Shell thickness deviation; and, (**c**) Elastic modulus deviation.

The stochastic analysis is implemented on the initial shell model, the 20th-step shell model and the 400th-step (optimal) shell model, respectively. The natural frequencies of all samples are solved by FEM using ANSYS. We first focus on the study of first natural frequencies, which are shown in Figure 8. Mean values and standard deviations of the first natural frequency are obtained as 4.56Hz and 0.15 Hz (initial), 6.13 Hz and 0.20 Hz (20th step), and 11.76 Hz and 0.36 Hz (optimal shape), respectively.

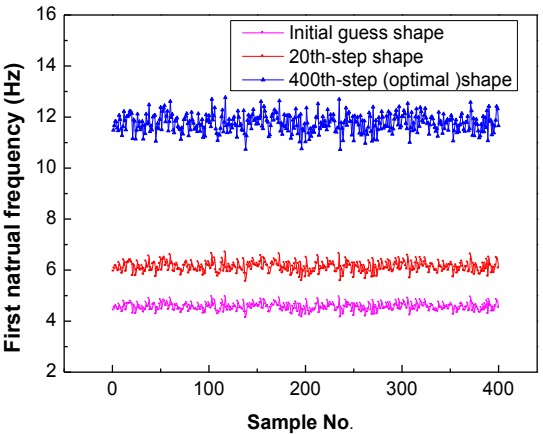

**Figure 8.** First natural frequency of Model 1.

Via the stochastic analysis, it is found that the first natural frequency follows a normal distribution. The cumulative distributions of first natural frequency of the three models are depicted and fitted in Figure 9 for clear analysis and comparison. As presented in Figure 9a, it can be seen that the value of first natural frequency of the initial guess ranges from 4.05 to 5.00 Hz and the cumulative distributions function (CDF) is fitted as

$$F(x) = \begin{cases} \frac{1}{1+e^{-10.97(x-4.55)}} & 4.05 \le x \le 5.00 \\ 0 & else \end{cases} \tag{7}$$

where $x$ denotes the first natural frequency. The determination coefficient is 0.9883, which shows a good approximation.

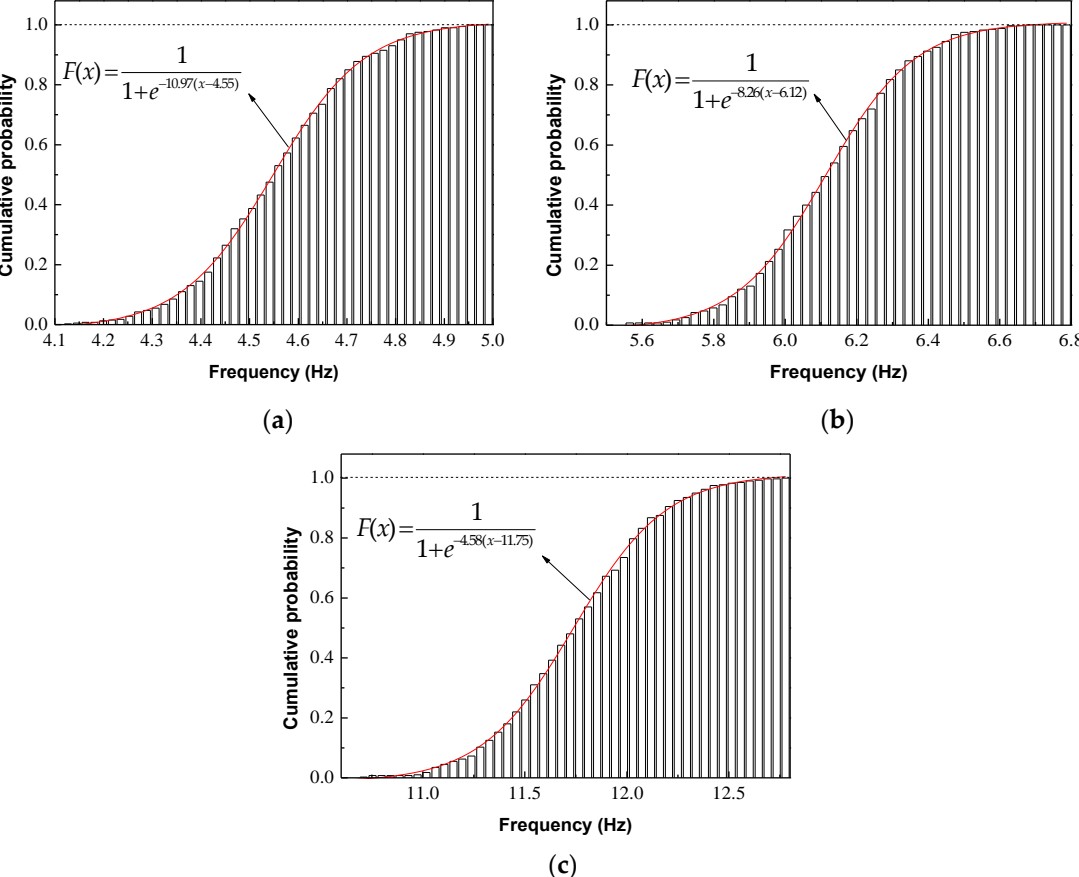

**Figure 9.** Cumulative distributions functions (CDFs) of the first natural frequencies of Model 1: (**a**) Initial guess; (**b**) 20th step; and, (**c**) 400th step (optimal).

It is also observed that the CDFs of 20th-step shape and optimal shape is similar to Equation (7) and fitted as Equations (8) and (9), respectively

$$F(x) = \begin{cases} \frac{1}{1+e^{-8.26(x-6.12)}} & 5.52 \le x \le 6.80 \\ 0 & else \end{cases} \tag{8}$$

$$F(x) = \begin{cases} \frac{1}{1+e^{-4.58(x-11.75)}} & 10.74 \le x \le 12.78 \\ 0 & else \end{cases} \tag{9}$$

Partial correlation coefficients with respect to the first natural frequency are calculated and subsequently normalized to rank the significance of each factor, as shown in Figure 10. It is illustrated

that, for initial guess, shell thickness and elastic modulus deviations impose relatively significant effects with similar weights, while the initial geometric imperfection is the least significant factor. In the case of the 20th-step shell, the significance ranking is similar to the initial guess, while the case of the optimal (400th-step) shape shows a difference that the effect of elastic modulus is remarkably higher than that of shell thickness deviation. The reason is that the increase (or decrease) of shell thickness leads to the increase (or decrease) of both stiffness and mass, which affects the frequencies in the opposite ways; in contrast, the increase (or decrease) of elastic modulus only affects the stiffness. Therefore, in most cases, the impact of the elastic modulus is larger than shell thickness. The difference between their significance depends on the geometrical shape to some extent.

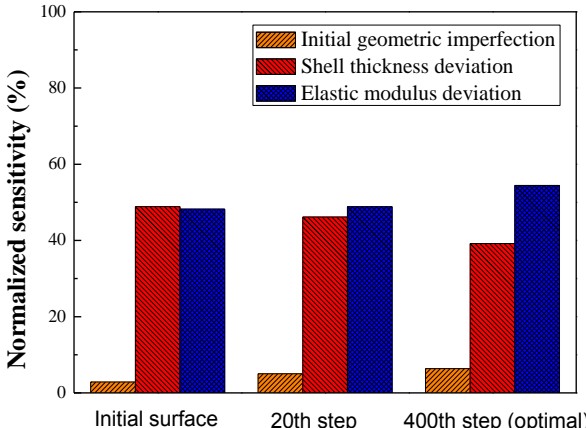

**Figure 10.** Normalized sensitivity of uncertainty factors of Model 1.

The stochastic and sensitivity analysis is repeated for the second and third natural frequency. Similar results are obtained.

### 4.2. Shell Model 2: A Free-Form Shell Consisting of Two Connected Positive—Gaussian Domes

#### 4.2.1. Model Generation

Figure 11 shows the shell of this example. The two long-span edges are simply supported and the other edges are free. The material properties are same to that in Table 2. The shape is generated by the proposed optimization approach. NURBS functions are adopted to represent the geometric shape. The control points are located, as shown in Figure 11a, which forms a $5 \times 3$ grid. Weight factors and the degrees of the basis functions are determined as similar to that in Example 1. The optimization aims at minimizing the strain energy under applied load, which is also determined as the uniformly distributed vertical load of $q = 3.5 \text{ kN/m}^2$. The triangular shell element is used with 1024 elements and 561 nodes.

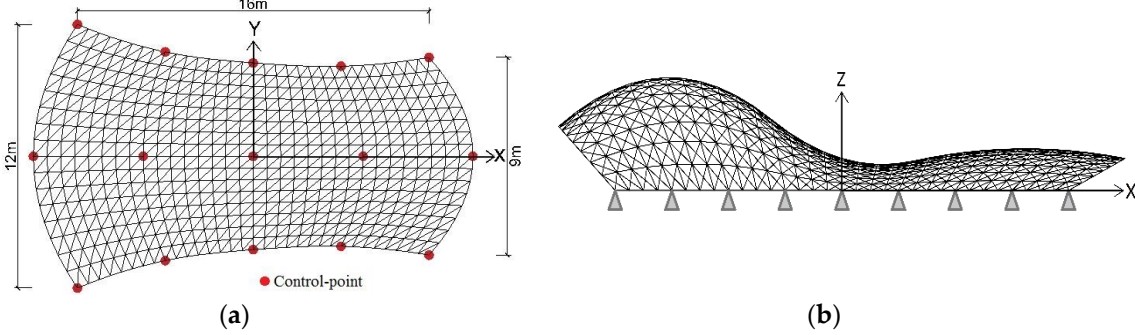

| (a) | (b) |

**Figure 11.** Layout of control points, mesh and the initial geometry of Model 2: (**a**) Layout of control points and mesh; and, (**b**) Initial guess shape.

As presented in Section 2, the gradient method is used to solve the optimization problem. The initial shape of this model is set up as a free-form shape, which consists of two connected positive-Gaussian domes. Similar to Example 1, the control point on the long-span edges is fixed, meaning that the location and shape of these two edges are not changing during the optimization process. The total number of optimization variables is 5, which is same to Example 1.

Figure 12 shows the convergence process of Model 2, which is slower than Model 1, since the shape is relatively complicated. The optimization converges at the 1000th step with a 65% decrease of strain energy. Figure 13 shows the changes of geometric shape of Model 2 during the optimization process. It can be seen that, since the control points on the edges are fixed, the general shape remain similar to the initial shape, which consists of two connected positive-Gaussian surfaces. However, the curvature increases overall.

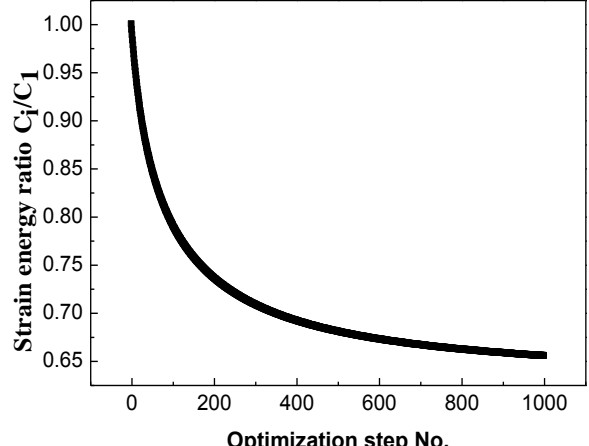

**Figure 12.** The convergence process of Model 2.

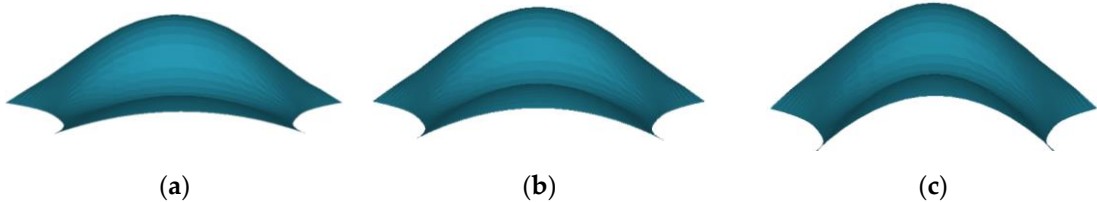

|      (a)      |      (b)      |      (c)      |

**Figure 13.** Shape variation of Model 2 during the optimization process: (**a**) Initial guess; (**b**) 100th step; and, (**c**) 1000th step (optimal).

Prior to the stochastic analysis, the deterministic analysis of free vibration is implemented on shell models with respect to several selected optimization steps, and Figure 14 depicts their first three natural frequencies. It is observed that, different from Model 1, both the first and third natural frequencies decrease through the optimization, while only the second natural frequency increases. It is also noted that the change of all the frequencies are slight. For example, the relative change of the first natural frequency is only 11.9%, which is significantly smaller than Model 1. This can be explained by the fact that, in this case, the stiffness is mainly increased by the increase of curvature, which results in the increase of mass. The natural frequencies are affected by both stiffness and mass, and they consequently show a slight change. The natural frequencies decrease when the increase of mass plays a more significant role than stiffness.

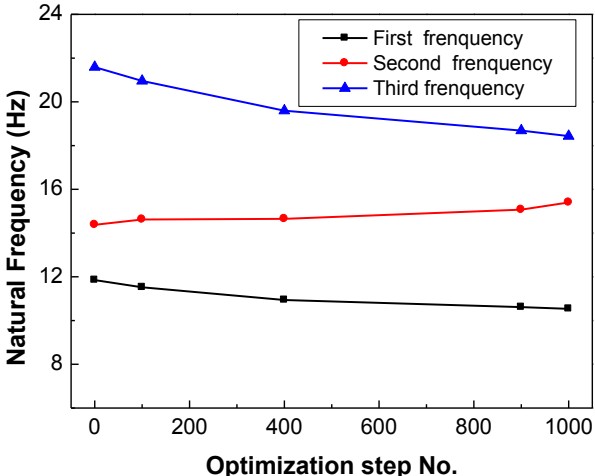

**Figure 14.** Evolution of the first three natural frequencies of Model 2 during the optimization process.

4.2.2. Stochastic and Sensitivity Analysis

The initial, 100th, and 1000th optimized shape of Model 2 in Figure 13 are selected to investigate the influence of uncertainty factors on free frequencies. The samples are generated by LHS. The sample size is also determined as 400, which has been checked for the current model by Equation (3), with $\xi_1, \xi_2$ as 0.01 and 0.001, respectively.

Firstly, the first natural frequency is studied, which is depicted in Figure 15. The mean values of the initial guess, 100th-step shape, 1000th-step shape is 11.91 Hz, 11.58 Hz, and 10.58 Hz, respectively. Their standard deviation is 0.28 Hz, 0.32 Hz, and 0.28 Hz, respectively. The stochastic analysis shows that the first frequency of this shell model always follows the normal distribution during the optimization process. Their CDFs are depicted in Figure 16 and fitted, which are similar to Model 1.

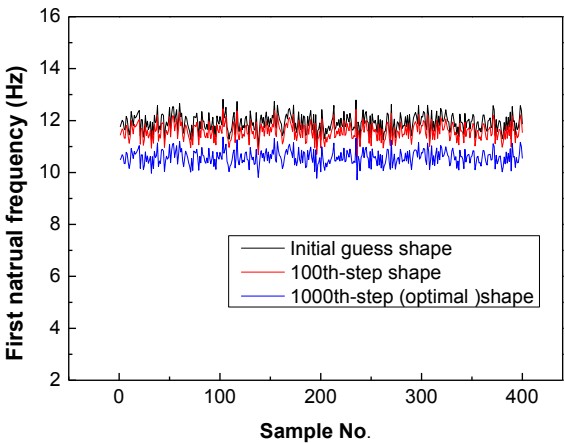

**Figure 15.** First natural frequency of Model 2.

Through the sensitivity analysis, the ranking of the three factors with respect to the first frequency is obtained, as shown in Figure 17. It can be seen that the elastic modulus deviation is the most important uncertainty, with a weight of around 60%; the shell thickness deviation has relatively small influence with a high weight of around 30%; and, the initial geometric imperfection imposes negligible effects with a weight of less than 10%. Generally, the ranking is similar to Example 1, except for that the sensitivity value of elastic modulus deviation is much higher. This could be attributed to the change of the mass, as discussed above.

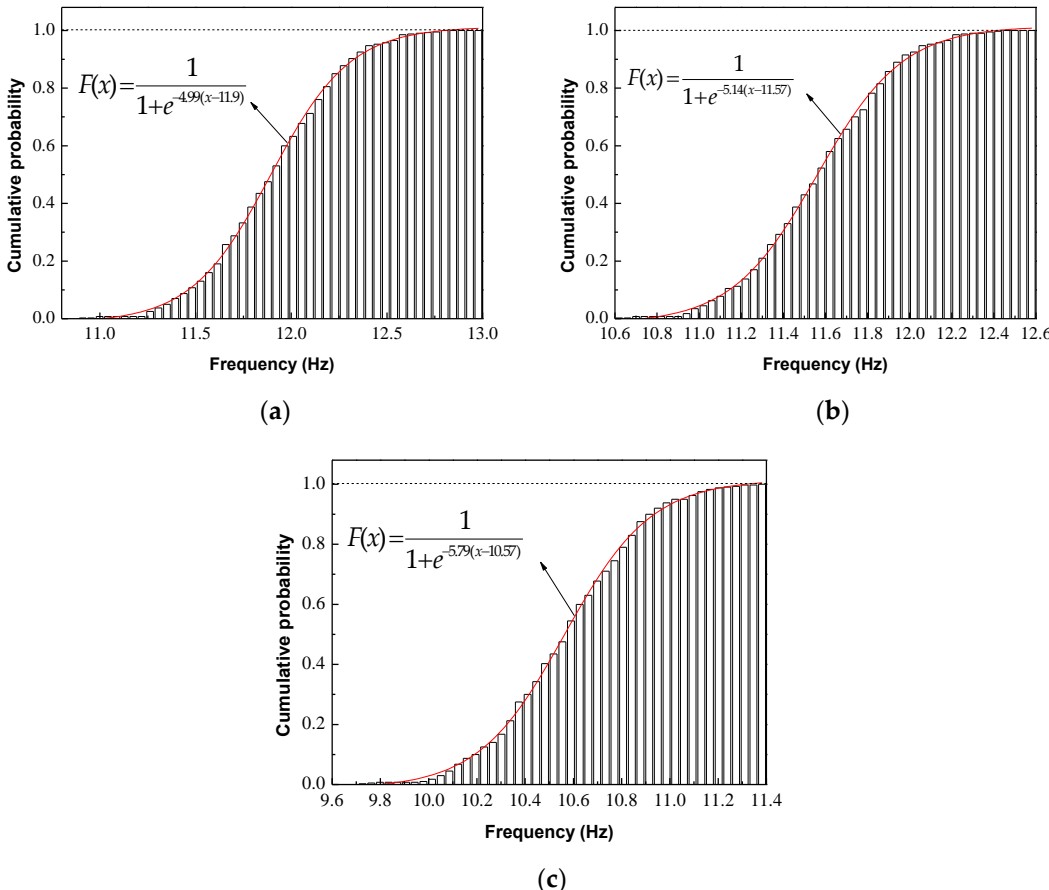

**Figure 16.** CDFs of the first natural frequencies of Model 2: (**a**) Initial guess; (**b**) 100th step; (**c**) 1000th step (optimal).

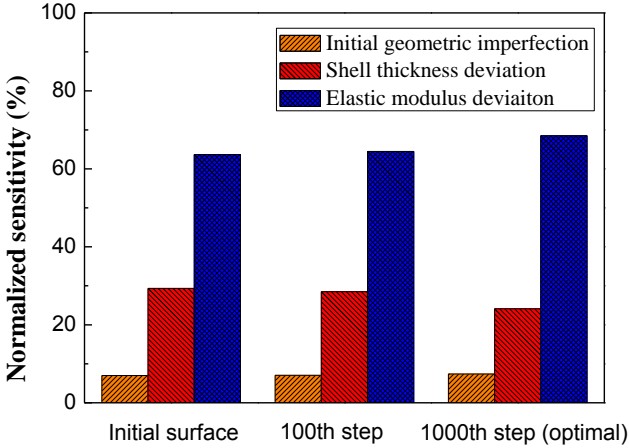

**Figure 17.** Normalized sensitivity of uncertainty factors of Model 2.

## 5. Conclusions

In this work, an optimization algorithm generates free-form shells. The stochastic analysis is carried out to investigate the stochastic characteristics of natural frequencies of free-form shells, in which three uncertainty factors are involved, including initial geometric imperfection, shell thickness deviation, and elastic modulus deviation. Furthermore, the global sensitivity analysis method is employed to rank the effect of three uncertainty factors. The main conclusions are drawn, as follows: (i) for the free-form shells investigated in this study, the natural frequencies of free-form shells follow a

normal distribution in term of the three uncertainty factors. The elastic modulus deviation imposes the greatest effect; shell thickness ranks the second; geometrical imperfection ranks the last with a much lower weight than the previous two factors. It is demonstrated that the free-form shells studied in this paper have good robustness on geometrical shape error; and, (ii) in the current study, the supported edges are fixed during the shape optimization, which lead to a limited shape change. Therefore, the stochastic characteristics do not significantly change during the optimization process. The case with variable edges needs further study.

**Author Contributions:** Conceptualization, B.S. and D.F.; Formal analysis, B.S., Y.M. and Z.X.; Methodology, B.S. and Y.M.; Software, L.Y.; Writing—original draft, B.S. and Z.X.; Writing—review & editing, B.S., Y.M. and D.F.; Funding acquisition, B.S.

**Funding:** This research was funded by the National Natural Science Foundation of China, grant number 51578211 and the Fundamental Research Funds for the Central Universities, grant number 2018B13814.

**Conflicts of Interest:** The authors declare no conflict of interest.

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
