# Peer review of "Stochastic Natural Vibration Analyses of Free-Form Shells"

_applsci, doi:10.3390/app9153168_

Round 1

Reviewer 1 Report

This article focuses on uncertainties influence on natural vibration characteristics of free-form shells. The steps to get to the proposed results are defined well and results are interestingly representing the effectiveness of the method.  

Author Response

RE: We thank the reviewer for the positive comments.

Reviewer 2 Report

In this article, the authors have generated two free-form shells and conducted stochastic analysis on evaluating the natural frequencies of the shells by considering just three uncertainty factors. The overall concept of the paper is interesting; however, the authors should make the following changes prior to the publication.

Comment 1:

The authors have considered just two free-form shells; however, they are generalizing their conclusions to any free-form shell (e.g., conclusion section line 324 through 328) which is not correct. In order to be able to generalize such a conclusion, many different free-form shells with different geometric and material characteristics (relating the elastic modulus) should be considered. Therefore, it is strongly suggested that the authors change those sentences in the abstract and conclusion sections in an appropriate way to convey that these conclusions are just for the two shells considered in this study.

Comment 2:

In line 134, the authors are claiming “ and  is generally determined based on previous experience or Knowledge”, it is not clear how? Based on what previous experience or knowledge?

Comment 3:

What is the justification of authors for considering equation (4) in the manuscript as the partial correlation coefficients between input uncertainty facto and output variables? How did they come up with this equation?

Comment 4:

In Table 1, row 1 which is corresponding to the initial geometric imperfections, why 2 numbers are stated?

Comment 5:

It is not clear how the weight factors of all control points are determined as 0.5 (Line 175). More explanation should be provided in this regard.

Comment 6:

The following typos should be corrected:

Line 55: “and” is missing between imperfection, deviations.

Line 58: Letter “s” in “leads” should be removed.

Line 64: LHS is mentioned in the introduction section for the first time. Thus, "Latin hypercube sampling" (LHS) should mention first, then the abbreviation "LHS" can be used.

Line 67: “:” should be used after "as follows".

Line 69: Passive form, "designed" should be used.

Line 74: Sentence does not convey meaning. "and" probably should be used between “spherical shells, cylindrical shells”.

Line 212: “Stand” should change to "Standard".

Figure 6: In the legend of Figure 6, "standare deviation" should change to "Standard Deviation".

Line 242: The word "ecoefficients" should change to "coefficients".

Line 308 (caption of Figure 16): Figure 16 is representing the CDFs of the first natural frequencies of Model 2. Thus, “Model 1” should change to “Model 2”.

Reviewer 3 Report

I think that the paper could be accepted in the present form. I noticed some typos or error in the english language. For example.

line 173 substitute liner with linear

line 248 the sentence "the reason being is that" sounds awkward to me

line 249 substitute effect with affects

line 250 substitute effect with affects

line 254 substitute The similar with Similar

Author Response

RE: We thank the reviewer for the overall positive comments. All above typos have been corrected. Meanwhile, the revised manuscript has been checked throughout to avoid additional typos.